# Work Potential and Work Performance during the First Try-Out of the Person-Centred Return to Work Rehabilitation Programme ReWork-Stroke: A Case Study

**DOI:** 10.3390/healthcare8040454

**Published:** 2020-11-02

**Authors:** Annika Öst Nilsson, Ulla Johansson, Elin Ekbladh, Birgitta Bernspång, Therese Hellman, Gunilla Eriksson

**Affiliations:** 1Department of Neurobiology, Care Sciences and Society, Division of Occupational Therapy, Karolinska Institutet, 14152 Huddinge, Sweden; annika.ost.nilsson@regiongavleborg.se (A.Ö.N.); uijohansson2@gmail.com (U.J.); gunilla.eriksson@ki.se (G.E.); 2Centre for Research & Development, Uppsala University/Region Gävleborg, 80188 Gävle, Sweden; 3Department of Health, Medicine and Caring Sciences, Linköping University, 58183 Linköping, Sweden; elin.ekbladh@liu.se; 4Department of Community Medicine and Rehabilitation, Umeå University, 90187 Umeå, Sweden; birgitta.bernspang@umu.se; 5Department of Medical Sciences, Occupational and Environmental Medicine, Uppsala University, 75124 Uppsala, Sweden; 6Department of Neuroscience, Rehabilitation Medicine, Uppsala University, 75185 Uppsala, Sweden

**Keywords:** rehabilitation, vocational, stroke, occupational therapy, work

## Abstract

Background: This case study explores changes in work potential and work performance for ten people who worked before their stroke while participating in the ReWork-Stroke programme. It describes measures performed by the occupational therapists to enhance work potential and work performance and the participants’ level of work re-entry nine months after the start of their work trial. Methods: Ten people who had experienced a mild or moderate stroke participated. Changes were assessed using the Worker Role Interview and the Assessment of Work Performance. Logbooks relating to work potential and work performance were analysed using content analysis. Results: The participants’ work potential was in general supportive to returning to work at baseline and remained so at the three-month follow-up. Most changes occurred in the environmental factors regarding the participants’ belief that adaptations at the workplace would make re-entry possible. Changes concerning work performance were predominately in a positive direction. Seven of the participants returned to paid work. Conclusion: The ReWork-Stroke programme seems promising for promoting changes in work potential, work performance, and return to paid work. However, further studies are needed to evaluate changes in work potential and work performance and the programme’s effectiveness for increasing work re-entry for people who have had stroke.

## 1. Introduction

Work plays an important role in peoples’ lives and is essential for their health [1]. Furthermore, work is closely connected to a person’s identity [2], and thus an important issue in rehabilitation. Not participating in working life after a stroke is a major consequence [3] that negatively impacts quality of life and leads to personal economic consequences [4], and brings high costs for society [5].

The incidence of stroke in those of working age is increasing [6]. About 20% of those having a stroke in Sweden are younger than the general retirement age of 65 years [6]. Approximately half of those at a younger age return to work [7], and those who do, continue to return to work for years post stroke [8]. Return to work has been described as a long process, encompassing a series of transitions and phases beginning at stroke onset and concluding when work re-entry or another satisfactory outcome has been achieved [9]. During the return-to-work process, several factors, interrelated and dependent of one another, influence the ability to work. These factors are, for example, health and functional capacity; competence; the individual’s attitudes and motivation towards work, as well as the content of the actual work and its organization; and the co-workers and management [10]. Changes in work ability are, according to Lederer et al. [11], a dynamic and relational phenomenon resulting from the interaction of multiple dimensions that overlap and influence each other. These interrelated factors are of importance for performing work tasks, as well as for the performance of other everyday occupations. Information regarding factors influencing work ability, such as work potential and work performance, is needed in order to understand how to support a return-to-work process.

Research on return-to-work programmes for people with stroke is scarce [12]. However, two small randomized controlled trials showed that return-to-work programmes after stroke, which use coordinators, were more effective as compared with usual rehabilitation [13,14]. For return-to-work programmes, recommended components include early contact with the workplace, tailored work tasks and workplace adjustments, involvement of the employer as well as co-workers, and work practice [12].

This study is part of a research project where a newly designed person-centred rehabilitation programme for return to work, ReWork-Stroke, was tried out in a rehabilitation context. The ReWork-Stroke was provided by an occupational therapist (OT) by measures to support the person that had experienced a stroke, to collaborate with other stakeholders involved and to coordinate the various measures in the return-to-work process. The components in the ReWork-Stroke programme are presented elsewhere [15]. There is limited knowledge regarding which measures can bring about change during a rehabilitation programme. Therefore, in the first test of the ReWork-Stroke programme, the aim of the present case study was to explore changes in work potential and work performance for ten people who worked before their stroke while participating in the programme. Furthermore, the aim was to describe measures performed by the occupational therapists (OTs) to enhance work potential and work performance during the programme, and the participants’ level of work re-entry nine months after the start of their work trial.

## 2. Materials and Methods

This study used a multiple case-study design [16] including participants who have had a stroke depicted in their daily contexts during their return-to-work (RTW) process. Both quantitative and qualitative data were used to explore the return-to-work process. A case-study methodology is useful when one wants to understand a real-life phenomenon in depth that is too complex for a survey or experimental design. Therefore, for this study, a multiple case design with two or more cases was chosen in order to cover different characteristics of the phenomenon [16]. The Regional Ethical Review Board, Stockholm, Sweden approved the study (2012/101-31/1).

### 2.1. Study Context

#### 2.1.1. Stakeholders in the Return-to-Work Process

The cases in this study are drawn from the Swedish context with its particular set-up and systems for the return-to-work-process. In Sweden, four stakeholders share the responsibility of supporting return to work. These stakeholders are the employer, healthcare staff, and representatives from the Swedish Social Insurance Agency (SSIA) and the Swedish Public Employment Service (in case the person does not have employment). The employer has a responsibility, in cooperation with the employee, to facilitate return to work through rehabilitation and workplace adjustments. This process is sometimes supported by a healthcare team/physician, while the SSIA has the overall responsibility for the administration of sick leave compensation during the return-to-work rehabilitation.

#### 2.1.2. The Intervention

The return-to-work rehabilitation programme, ReWork-Stroke, is based on existing knowledge [17,18,19] and is person-centred [20,21] but contains some generic elements [15]. Two OTs who were skilled in rehabilitation after a stroke, with clinical experience of 14 and 25 years, respectively, coordinated the return-to-work programmes. They were part of specialist brain injury teams at two different hospitals in the middle of Sweden.

The programme started with a preparation phase of varying length (reported in Table 1), where the client´s family and living situation, routines in daily life, as well as resources and obstacles for going back to work were mapped out. The prerequisites for return to work and the client´s goal for the rehabilitation were discussed together with the participant. Information on stroke and work-related rehabilitation was provided to the employer, co-workers, close relatives, and other involved stakeholders. A plan concerning a work trial and time for follow-up were elaborated at the client´s workplace in cooperation with the other stakeholders involved. This plan included the start time for the work trial, work tasks to start with, and routine for feedback and support. During the work-trial phase, the OTs made regular visits to participants’ workplaces, i.e., twice a month during the period of the three-month work trial, to support the participants and their employers/managers/co-workers. Individual advice and solutions were discussed, regarding how to handle the consequences of a stroke in situations at work, for example, the OTs contacted the stakeholders involved regularly and information on the rehabilitation process was continuously exchanged.

Collaboration between all stakeholders was encouraged throughout the entire process. An evaluation of the participants’ work ability was performed and the plan for return to work was adjusted according to the individuals’ needs and conditions, and the opportunities at the workplace.

### 2.2. Participants

People eligible for inclusion were those who had experienced a stroke and who (1) were referred for rehabilitation to specialized brain injury rehabilitation units in two cities in Sweden, (2) were 20–63 years of age, (3) worked before stroke, (4) wanted to return to work, (5) were estimated by the rehabilitation team to be in the phase of return-to-work rehabilitation, and (6) had the ability to communicate in Swedish. Exclusion criteria included the following: diagnosed with dementia or diagnosed with other neurological or psychiatric disorders. The present study was based on ten participants who received rehabilitation at the specialist brain injury units. They were consecutively included and completed the return-to-work programme, including a three-month follow-up and a call concerning work re-entry at nine months after start of the work trial. Demographic characteristics of the participants are presented in Table 1. Stroke severity at onset was calculated based on ability in daily life activities (ADL) and the Barthel index was used with the categories mild stroke (50–100), moderate stroke (15–49), or severe stroke (<14) [22]. The participants consequences of stroke were retrieved from the medical records

### 2.3. Data Collection

Multiple data sources were used to collect information sufficient to describe changes during the work trial phase in the ReWork-Stroke programme. The data comprised information on work performance, work potential, fatigue and perceived impact of stroke, and information in logbooks on the measures taken by the occupational therapists. Data were collected at the preparation phase (baseline) and at follow-up, which took place three months after the start of the work trial. Two data collection phases were used in order to detect changes during the return-to-work process. The baseline information was also used to target specific challenges to deal with during the work trial. Logbooks were continuously written by the OTs regarding measures they provided in the rehabilitation for each participant during the whole programme. Data from the medical records regarding activities of daily living (ADLs) and bodily functions were collected at baseline and used to describe the sample. The Assessment of Work Performance (AWP) and the Worker Role Interview (WRI) were conducted by two independent vocational specialists from the Swedish Public Employment Office not involved in the execution of the RTW programme. These specialists were skilled in conducting the assessments and interpreting the results. The OTs providing the ReWork-Stroke had access to the interpretation of the results of these assessments as a basis for their interventions during the rehabilitation. The Fatigue severity scale (FSS) and the Stroke impact scale (SIS) were administrated by the OTs. Information on level of work re-entry was collected in follow-up calls conducted by the OTs, nine months after the start of the work trial.

#### 2.3.1. Instruments

Work performance, defined as the individuals’ ability to satisfactorily handle and carry out different work activities and tasks, was measured using the AWP. The aim of the AWP is to identify clients’ work skills by observing how efficiently and appropriately the client performs a work task. The AWP is theoretically founded on the Model of Human Occupation and assesses clients’ work performance in the following three domains: motor skills, process skills, and communication and interaction skills [23]. The items in the motor skills and process skills domains are presented in Table 4. The domain, communication and interaction skills, consists of four items, but none of these were used. In this study, the AWP assessments were conducted in a constructed environment where the assembly and the administrative tasks, two structured work tasks developed for the AWP assessments, were used. The assembly task was comprised of assembling shelves and placing materials on them in a logical order. The administrative task was comprised of registering orders of nameplates in an Excel file. The performance on each work task was scored on a Likert scale where 1 = incompetent performance, 2 = limited performance, 3 = questionable performance, and 4 = competent performance. The AWP has presented good psychometric properties [24,25,26].

Work potential was measured using the Swedish version of WRI-S. The WRI is also theoretically founded on the Model of Human Occupation and aims to identify psychosocial and environmental factors that influence a person’s ability to return to work after injury or illness. The WRI interview, conducted by using a non-standardized semi-structured interview guide, is designed to collect data concerning the content areas of motivational factors, lifestyle factors, and environmental factors [23]. After the interview, a therapist-administered rating scale is used to assess the client’s work potential in 16 items (see Table 3) with standardized definitions, each item corresponding to one of the content areas. The items are assessed on a Likert scale where 1 = strongly interferes with, 2 = interferes with, 3 = supports, and 4 = strongly supports the client’s possibilities of returning to work. Psychometric properties of the Swedish version of the WRI have been shown to be sound [27,28,29] and the WRI has been found to be a clinically useful and person-centred assessment [30].

Fatigue was assessed using the self-report instrument FSS-7 item version (FSS-7). This is a valid instrument for evaluating changes in fatigue over time among people that have had a stroke. The final score is the mean of the seven items graded from 1 (strong disagreement) to 7 (strong agreement) with having fatigue. The cut-off score used for the presence of fatigue post stroke is a mean score of 4 [31].

Perceived impact of stroke. SIS 3.0 was used to assess perceived impact of stroke. The SIS 3.0 consists of 59 items representing eight domains, i.e., strength, memory and thinking, emotions, communication, activities of daily living (ADL)/instrumental activities of daily living (IADL), mobility, hand function, and participation [32,33]. The person who has had a stroke scores the items on a scale from 1 to 5. By using an algorithm, aggregated scores for each domain are calculated. The domain score ranges from 0 (maximum perceived impact of stroke) to 100 (no impact) [32]. The SIS includes one further item regarding the perceived global recovery after stroke, rated on a visual analogue scale ranging from 0 (no recovery) to 100 (full recovery). The psychometric properties of the SIS have been shown to be good [32,33,34].

#### 2.3.2. Logbooks

The OTs documented the content of the intervention for each participant in the rehabilitation programme. These logbooks were continuously written during the preparation and work trial phases. The material comprises, in total, about 100 pages of written text. Data from the logbooks were expected to give an understanding of what happened in the cases during the participation in the ReWork-Stroke regarding their work potential and work performance, in relation to the measures performed by the OTs in the programme. These measures could, for example, be the use of strategies and adjustments such as working hours, work environment, as well as cooperation with others.

### 2.4. Data Analysis

The data analysis was performed by the authors in two steps. First, all data from the instruments were analysed. Thereafter, data from the logbooks were analysed with a focus on additional information related to the performed measures in the programme in order to describe changes for the participants during their return-to-work process. Descriptive statistics were used for the demographic data. Aggregated domain scores were generated for each SIS domain. Clinically meaningful changes in SIS domain scores were considered as positive (+15 points or more), negative (−15 or lower), or no change with a difference between −14 and +14 [33]. In the analyses of work potential (WRI) and work performance (AWP), data were analysed on an individual level to describe changes for each participant. Concerning the WRI, a positive change to a supporting item was identified when an item rating of 1 or 2 (interfering) had changed to a rating of 3 or 4 (supporting) at follow-up, and negative change was identified when the item rating had changed in the opposite way. Concerning the AWP, a positive change to a competent performance was identified when the rating of the performance between baseline and after three months of work trial had reached 4 on the actual item. A negative change was identified when the rating had decreased into a lower rating. Details in the changes in ratings in WRI and AWP are reported in Table 3 and Table 4 and explained in footnotes. On the basis of these changes, three cases were identified for in-depth description of their return-to-work process during the ReWork-Stroke programme.

A manifest deductive content analysis [35] was applied in the analysis of the logbooks for the three cases chosen. Firstly, the text in the logbooks was read through by three of the authors (U.J., G.E., A.Ö.N.) to become familiar with the OTs’ information. Thereafter, the unit of analysis was identified based on the factors of motivation, lifestyle, and environment originating from the WRI, and the domains of motor and process skills originating from the AWP. In the next step the text was divided into meaning units, which considered capturing content in the factors and domains. Then, the meaning units were condensed and coded [36] into words close to the OTs’ text. The codes describing the OTs’ measures from the three logbooks were ordered, consistent with the different factors and domains in WRI and AWP, to be used in the three case descriptions. During the analysis process, the authors (UJ, GE) moved back and forth to compare the text in the logbooks and the text in the emerging analysis.

## 3. Results

In this study all participants reached changes in work potential and work performance, of which the majority were positive during the ReWork-Stroke, shown in Table 3 and Table 4. The majority of the participants had returned to part-time work nine months after the start of the work trial. Details on work re-entry are reported in Table 5. A variety of the OTs’ measures were identified in three case descriptions. These measures supported the participants in their individual work situation and contributed to enable the positive changes.

Time from inclusion and start of a work trial varied from 4.5 to 19 months due to different circumstances, such as difficulties in finding suitable work tasks that fitted the participant’s cognitive limitations (see Table 1). The impact of stroke among the participants was perceived as relatively similar while they were in the programme. The stroke impact scale domains of emotions, participation, and stroke recovery were affected (domain score about 60) at baseline for nearly all participants and continued to be affected at follow-up. The motor domains were affected for some participants, as well as memory, communication, and ADL/IADL (See Table 2).

### 3.1. Work Potential

Changes in work potential between baseline and after three months of the work trial are presented in Table 3. For the majority of the participants, all three factors were supportive for their abilities to return to work as early as when starting their work trial. To a great extent, these factors remained supportive at the three-month follow-up. Half of the 10 participants had a sustained supportive factor in 14 out of 16 items in WRI from baseline to the three-month follow-up. Additionally, two participants had a sustained supportive factor in 13 of the 16 items. Eight of the participants had one to six positive changes in work potential during their participation in the work trial in ReWork-Stroke. Six of the participants had one or two negative changes during this time period.

### 3.2. Work Performance

Changes in work performance between baseline and after three months of work trial are presented in Table 4. All participants reported a positive change to a competent performance on one or more items. Six participants had positive changes on one or more items but did not reach a competent performance. Changes were predominantly in a positive direction from incompetent to competent performance for both motor and process skills. Still, four participants had negative changes. Nine participants had competent performance both at baseline and after work trial on several items.

### 3.3. Work Re-Entry

Three participants reported that they were still on work trial nine months after the beginning of the work trial, while seven participants had returned to paid employment to some extent. The self-reported work re-entry levels among the ten participants, nine months after the beginning of the work trial, are presented in Table 5.

### 3.4. Case Descriptions

#### 3.4.1. Participant E: Elisabeth

Elisabeth was nearly 60 years old and lived with her husband when she experienced a mild stroke. She worked as a manager at the time of stroke and the organization at the workplace was in the middle of a large change. Elisabeth was included in the ReWork-Stroke programme 8.5 months after her stroke. Due to her stroke, she had difficulties with reading, writing, fatigue, and ability to concentrate. These consequences after her stroke made it difficult for her to remain in a management position, which is why another person took this role. Elisabeth enjoyed being back at the workplace again and was looking forward to her return to work. She was supported by one of her colleagues and they had a close cooperation with the OT who came to the workplace regularly.

The WRI scores at baseline showed that some items concerning motivation for work were interfering with the return to work, i.e., expectation of job success, commitment to work and work-related goals, and they did not improve during the work trial. The logbooks revealed that Elisabeth was unsure about her possibilities to return to work. Her concerns about how to manage work tasks were discussed several times. Strategies on how to deal with the consequences of her stroke were suggested by the OT. The other work potential factors were mostly supportive or became supportive during the work trial. The AWP showed that Elisabeth had positive changes on all motor items between baseline and follow-up and reached competent performance on all except physical energy. However, on the process items, there was no change in three of the five items, i.e., knowledge, temporal organization and adaptation, and competent performances were not achieved.

Different strategies and aids were proposed and discussed during the work trial and were also successively implemented and used by Elisabeth. Strategies were, for example, tape recording of meetings to substitute for memory problems, use of a speech dictation device instead of writing minutes, writing down of main points before calling someone, use of to-do lists to get structure and support for memory, and taking small breaks to stay more alert. Technical aids that facilitated her work were installed on her computer.

During the work trial, ongoing discussions were conducted to find suitable work tasks that were manageable for her and that could be paid for in the long run. For example, she tried out working with bookkeeping as numbers were easier for her to work with than words. These tasks worked out well for her but were very tiring. She also tried some of her ordinary work tasks such as planning for and leading meetings.

It also became evident to co-workers during the work trial that Elisabeth had some difficulties to remember information. This was an interfering factor at work as she could, for example, forget to do work tasks that should have been done before meetings.

Discussions about her possibility to return to her former job duties despite her aphasia continued during the work trial.

At the follow-up nine months after the start of the work trial, Elisabeth was still in a work trial.

#### 3.4.2. Participant G: Gina

Gina was a 52-year-old administrator who was included in the ReWork-Stroke programme 19 months after stroke onset. On her inclusion, she still had some physical impairment, as well as aphasia and she also described an impact on cognitive functions such as concentration.

Her WRI scores indicated a difficulty in working out abilities and limitations and this was one area that, according to the logbook, has been central during the work trial. Together with Gina and her employer, the OT discussed and planned for work tasks, working hours, and the social environment.

At the end of the work trial, the WRI showed both some positive and some negative changes. Most of the positive changes were for motivational factors, while negative change occurred for lifestyle factors that addressed daily routines. The logbook indicated that work demanded a lot of energy for Gina, which affected her life outside work.

AWP showed a positive change in motor items with the exception of physical energy, which was below a level supporting return to work at the end of the work trial. This was, as underpinned in the notes of the logbook, an indication of the balancing between her strong wish to return to work and her energy to do so. During the OT’s recurrent visits to the workplace, subjects such as the pace of expanding working hours were often on the agenda. Gina wanted to hasten the process, but her boss and the OT were helping her to take it step-by-step by evaluating and discussing her achievements.

The other motor and process items showed unaltered competency or change to competent performance after the work trial.

Gina was working 50% at follow-up nine months after start of the work trial.

#### 3.4.3. Participant H: Henry

Henry was 50 years old when he experienced a moderate stroke. He was married, had two children, and worked as a craftsman. After his stroke, he had both physical and cognitive difficulties. His balance was affected as well as his memory and vision. Due to his vision impairment, he was not allowed to drive which was a concern when discussing returning to work and the bus option was mentioned. The time from stroke onset to inclusion in ReWork-Stroke was seven months. Henry realized that his work tasks at his former workplace were too difficult due to the consequences of his stroke. He therefore used his contacts and found another workplace where he could start his work trial.

It was mainly the motivational factors, lifestyle factors and environmental factors in the WRI that supported his return to work. However, according to the WRI assessment Henry was unsure about his abilities and limitations after his stroke. This was also a focus during the work trial where his boss collaborated with Henry and the OT to find suitable work tasks for him. During the work trial, he could try out both his severity of work tasks and working hours.

The AWP scores showed a change to competent performance on three motor items. This was an important improvement, as Henry had physically demanding work. The AWP process items were all scored as incompetent performance after three months of the work trial. Two of the items, temporal organization and organization of space/objects, had changed from competent to incompetent. Strategies to handle the consequences after the stroke at work were discussed and suggested by the OT during the three months of the work trial. Henry was recommended to take short breaks and to only focus on one task at a time. His co-workers were supportive and assisted in reminding him about taking breaks and to adjust his working pace (work slower). Henry also needed to use different strategies and techniques to reduce the impact of impaired memory, such as to-do lists and the use of a calendar, both at work and in household chores at home.

Henry tried to work full days and physically it worked out all right. He even had the energy to walk the dog after work, but his memory problems became worse. In cooperation with the OT and the workplace, the schedule was changed so that Henry could have a day of rest between two working days. The WRI item “pursues interest”, showed that he had enhanced his ability to seize opportunities to make his life stimulating and meaningful both in and outside work.

At the follow-up nine months after the start of the work trial, Henry was still on work trial.

## 4. Discussion

The findings in this study identified a variety of changes in both work potential and work performance, and the majority of these were positive. Still, the case study revealed that the changes were individually influenced without any clearly emerging pattern among the participants. Furthermore, the measures taken by the OTs varied to a great extent, which implicate that a person-centred approach in a return-to-work programme for people who have had a stroke is needed. The findings will be further discussed below.

To the best of our knowledge, exploring changes in work potential and work performance during a return-to-work programme by using assessment instruments, in combination with documentation from logbooks, has not been done before. Findings from this study revealed that many work potential items were assessed as supportive in the preparation phase and remained supportive. One reason for some of these supportive WRI scores was probably because willingness to return to work and being estimated as ready to start the return-to-work process were inclusion criteria for the project. Still, the motivational factors in the WRI were the most frequently reported items as being interfering factors, both at baseline and at follow-up. This indicates that the motivational factors are difficult to target in a return-to-work programme. Even so, it is highly necessary to reflect on these results, as previous research has found that being motivated to work was of great importance for returning to work for people who have had a stroke [36]. The fact that, in some cases, the motivational factors were interfering may have reflected the participants’ growing awareness of their new situation and how the consequences of a stroke impacted their abilities to perform work tasks and, by extension, their opportunities to return to their former workplace and working life per se. These results are in line with the findings from Braathen et al. [37], who described the process their participants went through when learning about changes in work ability and improving coping strategies in interaction with the workplace. Still, this needs to be investigated further in future research.

The perceived impact of stroke, measured with SIS, was rather similar between the measuring points, which is not surprising as more than a year had passed since stroke onset for seven of the participants. However, in the domain of emotions, most of the participants reported clinically meaningful changes. These changes might reflect the fact that beginning the work trial was beneficial to their mood. Earlier studies [38,39,40] have highlighted the importance of employment for quality of life. The clinically meaningful changes in the domain participation for four of the participants might indicate a positive experience of having returned to work and being back in the social context at work.

The differences in the results of the AWP assessments between the participants illustrate that the consequences of a stroke can influence work performance in several ways. Different patterns were also found among the participants in the WRI, which highlights the complexity of work ability [10]. The decision to return to work or not cannot be explained by any single factor in this sample. As revealed in the logbooks, individualized support was needed to meet the heterogeneity in the sample, i.e., stroke severity, cognitive as well as motor impairments, time since onset, time for preparation before start of work trial, varying social support, distances from home to workplace, and profession. The OT suggested various strategies for dealing with the consequences of a stroke at the workplace. The variety of strategies used, and the solutions developed in collaboration at the workplace are in line with a person-centred approach [20,21] and might entail the positive changes in work potential and work performance. Furthermore, the findings, in the present study, revealed that the experienced OTs supported participants in their specific work situation. Thereby, the workplaces became natural platforms for discussions between the participants and other stakeholders on strategies and adaptations needed. This feature has previously been highlighted as very important [41].

Interestingly, it became evident in the present study that it was not enough to discuss strategies to cope at work, as the logbooks revealed that strategies targeting lifestyle balance [42] were also discussed. These results might be understood in relation to the results of the WRI, which identified that there were still a few interfering factors within the area of lifestyle factors at follow-up, despite the fact that the new routine “to work” was initiated. The importance of focusing on a variety of factors regarding perceived work ability has also been reported elsewhere [10].

### Methodological Considerations

This study used a multiple case-study design to obtain in-depth information about work potential and work performance during a return-to-work process. Both quantitative and qualitative data were used to illuminate the return to work process.

The use of field notes from logbooks added examples on measures during the programme, and thereby complemented the quantitative data. This gave an increased understanding concerning work performance and work potential on an individual level. The underpinning of this choice was the complexity of the phenomenon, i.e., return to work after a stroke. As described in the literature, work ability consists of and, relies on, many factors that are personal, workplace related, and environmental [10]. A case study design is suitable when studying different characteristics in-depth [16]. The more detailed presentation of three cases was intended to give examples of different situations. It also demonstrates the variety of factors impacting a successful return to work after a stroke. One strength of the study was that the participants represent different professions, educational backgrounds, living areas, and family situations. Therefore, we could ascertain the richness of the data illustrating the phenomenon from different aspects.

In this study, the AWP and WRI assessments of work ability were performed by two vocational specialists from the Swedish Public Employment Office, who were skilled in using these assessments and not involved in the rehabilitation of the participants, which was one strength of the study. The AWPs were performed in a constructed setting at a clinic. It would have been interesting, as a complementary aspect to the study, to perform AWPs at the participants’ specific workplaces with their real work tasks, as has been proposed by Karlsson and co-authors [25]. In that case, the information from the assessment would probably have been more directly connected to the challenges at the workplace. The high scores on AWP might be because the majority had experienced a mild stroke. One limitation was that communication and interaction skills in AWP were not assessed, as each participant was assessed separately. Thus, interaction with other individuals was not possible. Other limitations to this study were the sample size that did not allow any generalization from the results and the relatively short time that the participants were followed in their return-to-work process. A longer follow-up period might have contributed valuable information concerning changes in work potential and work performance, as the ability to work might change, even after several years [8]. The inclusion criteria to be motivated to return to work is reflected in scores on WRI area Motivational factors. This probably affects the potential for positive change in scores during the work trial period.

The OTs, in this study, had extensive knowledge and experience of stroke and vocational rehabilitation, which was important to be aware of and to be considered when providing the ReWork-Stroke programme. They were familiar with providing individual support in the work environment connected to specific situations. In this study, they were able to increase our understanding about measures in the programme that could bring about changes in work ability and return to work after a stroke.

## 5. Conclusions

Changes in work potential and work performance, of which the majority were positive, occurred for the participants while taking part in the ReWork-Stroke programme. It became evident that the changes occurred in various domains in line with the functional and activity limitations of the participants. On the basis of this variety, it was also clear that the OTs’ measures varied across cases, but all had a close collaboration with the manager and colleagues at the workplace. In this study, seven of ten persons had paid employment to some extent nine months after starting the work trial. This is, however, the first study on the ReWork-Stroke programme and further studies are needed to evaluate changes in work potential and work performance and the programme´s effectiveness in increasing work re-entry for people who have experienced a stroke.

Important implications drawn from this study are, thus, the importance of gathering information regarding work potential and work performance when planning the measures taken. Another important conclusion is that the measures need to be planned based on the patients’ abilities, the demands of the workplace, as well as the patients’ life situation. When taking all these factors into account, a process of co-creating work ability might occur by means of the person-centred approach in the return-to-work programme and by means of the collaboration with all involved at the workplace.

## Figures and Tables

**Table 1 healthcare-08-00454-t001:** Demographic characteristics, information on stroke severity and consequences, and time for inclusion in ReWork-Stroke for the participants (*n* = 10).

Participants	A	B	C	D	E	F	G	H	I	J
Age at stroke onset	40	52	42	44	57	55	52	50	54	48
Gender	Female	Male	Male	Male	Female	Female	Female	Male	Male	Male
Civil status	Single	Married	Married	Married	Married	Married	Married	Married	Single	Single
Children living at home		1 child	3 children	1 child				2 children		
Profession	Instructor	Transport organizer	Craftsman	Social service person	Manager	IT advisor	University admin	Craftsman	Craftsman	IT advisor
Consequences of stroke	Fatigue Vision Aphasia	Fatigue	Fatigue Memory	Fatigue Memory Attention Balance Dizziness	Fatigue Attention Vision Aphasia	Fatigue	Attention Vision Aphasia Muscle weakness in one arm	Memory Vision Balance	Muscle weakness in one leg	Fatigue Muscle weakness on one side of the body
Stroke severity (BI)	Mild	Mild	Moderate	Moderate	Mild	Mild	Mild	Moderate	Mild	Mild
Onset to inclusion, months	14	4.5	7.5	7.5	8.5	8	19	7	5.5	9.5
Inclusion to start of work-trial, months	4	2	3	8,5	3	5	5	8	3	5
Fatigue severity scale, mean (1–7), baseline/3 months	4.1 */4.1 *	1/1.9	6 */4.7 *	5.4 */4.7 *	6.4 */5.9 *	5.7 */6.6 *	1.9/1.4	3.1/2.3	3.1/3	4.3 */5.9 *

* Indicates a score above the cut-off score for severe fatigue = 4. BI, Barthel index; IT, information technology.

**Table 2 healthcare-08-00454-t002:** Impact of stroke at the baseline and after three months of the work trial in the stroke impact scale 3.0 (SIS) domains, and clinically meaningful changes between these phases (*n* = 10).

SIS Domains	Participants									
	A	B	C	D	E	F	G	H	I	J
Strength		100/-		75/75		75/75	50/50		**44/63**	38/44
Memory	93/100	82/89	68/68	75/75	100/96	50/54		**46/82**		100/96
Emotions	**61/78**	**75/97**	**61/92**	**56/83**	**61/78**	**64/94**	**61/78**	**53/72**	61/75	**58/75**
Communication	69/79	69/79	97/96	91/93	50/57	75/64	91/89	81/89		97/96
ADL/IADL			95/100	88/95		88/75	65/78	85/98	100/98	63/63
Mobility				**83/100**		100/89	97/94	97/100	89/97	61/75
Hand function		100/-		90/100		100/89	0/10			15/20
Participation	86/84	**71/91**	57/57	**61/97**	61/72	54/50	79/91	**46/100**	79/91	**39/91**
Stroke recovery	61/67	97/85	60/60	71/85	**65/50**	70/60	30/40	92/86	50/60	70/80
**Participants**	**A**	**B**	**C**	**D**	**E**	**F**	**G**	**H**	**I**	**J**
Total change based on all SIS domains:	P:1	P:2	P:1	P:3	P:1	P:1	P:1	P:3	P:1	P:2
	N:0	N:0	N:0	N:0	N:1	N:0	N:0	N:0	N:0	N:0

P, positive clinically meaningful change; N, negative clinically meaningful change; empty spaces, a domain score of 100 at both data collection points. Bold numbers indicate clinically meaningful changes between baseline and three months of work trial.

**Table 3 healthcare-08-00454-t003:** Participants’ changes in work potential between baseline and after 3 months of work trial assessed with the worker role interview (WRI).

Content Area	WRI Items	Participants										
		A	B	C	D	E	F	G	H	I	J	
MOTIVATIONAL FACTORS Personal causation, item 1–3 Values, item 4-5 Interests, item 6–7	Assesses abilities and limitations		P			N	P	I	P			
	Expectation of job success			N		I		P				
	Takes responsibility							P				
	Commitment to work					I		P				
	Work-related goals					I		P				
	Enjoys work							I				
	Pursues interests			I				I	P	N		
LIFESTYLE FACTORS Roles, item 8–9 Habits, item 10–12	Appraises work expectations											
	Influence of other roles			P								
	Work habits			−		−						
	Daily routines				P			N				
	Adapts routine to minimize difficulties			P		N		N				
ENVIRONMENTAL FACTORS Workplace related items, 13, 15, 16	Perception of work setting		P	I	P	P		P				
	Perception of family and peers	P				P	N			−		
	Perception of boss	I				I		P			N	
	Perception of co-workers			P	P	P		I	P		N	
	Total change based on all WRI items:	P:1	P:2	P:3	P:3	P:3	P:1	P:6	P:3	P:0	P:0	
		N:0	N:0	N:1	N:0	N:2	N:1	N:2	N:0	N:1	N:2	

P, positive change in actual work potential factor between baseline and follow-up, i.e., change from interfering (score 1,2) to supporting (score 3,4); N, negative change in actual work potential between baseline and follow-up, i.e., change from supporting (score 3,4) to interfering (score 1,2); I, still an interfering factor for the participant’s work potential, i.e., no change between baseline and after 3 months of work trial; empty space, still a supporting factor for the participant’s work potential, i.e., no change between baseline and after 3 months of work trial; −, missing data.

**Table 4 healthcare-08-00454-t004:** Participants´ changes on assessment in work performance between baseline and after three months of work trial, assessed with the Assessment of Work Performance (AWP).

AWP Items	Participants									
	A	B	C	D	E	F	G	H	I	J
Posture	B	B		P	P	N	P	P		P
Mobility		P		P	P		−	−		P
Coordination				P	P		MP	P		P
Strength					P		P			P
Physical energy	P	N	B	P	MP	N	B	P	B	P
Mental energy	MP	P	P	P	MP	B		B	P	P
Knowledge		N	MP	P	B		P	B		
Temporal organization		N	P	P	B		P	N		
Organisation of space/objects	N	B	P	MP	MP	P		N	P	
Adaptation		MP	P	MP	B	P	P	B	P	
Total change per person based on all items	P = 1	P = 2	P = 4	P = 7	P = 4	P = 2	P = 5	P = 3	P = 3	P = 6
	MP = 1	MP = 1	MP = 1	MP = 2	MP = 3	MP = 0	MP = 1	MP = 0	MP = 0	MP = 0
	N = 1	N = 3	N = 0	N = 0	N = 0	N = 2	N = 0	N = 2	N = 0	N = 0

P, positive change in AWP item, reached a competent performance between baseline and after three months of work trial; MP, positive change in AWP item, but not reached a competent performance between baseline and after three months of work trial; N, negative change in AWP item between baseline and after three months of work trial; B, no change (same) in AWP item between baseline and after three 3 months of work trial and below competent performance; empty space, competent performance both at baseline and after three months of work trial; −, missing data.

**Table 5 healthcare-08-00454-t005:** Levels of work re-entry at 9 months after start of the work trial.

Participants	A	B	C	D	E	F	G	H	I	J
Work status	In work trial	75%	50%	25%	In work trial	50%	50%	In work trial	75%	25%

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
