# Peer review of "Work Potential and Work Performance during the First Try-Out of the Person-Centred Return to Work Rehabilitation Programme ReWork-Stroke: A Case Study"

_healthcare, 2020, doi:10.3390/healthcare8040454_

Round 1

Reviewer 1 Report

This is an interesting study aimed to investigate the changes in work potential and work performance for people who participated in the ReWork-stroke.

The present study is relevant to propose a newly return to work programme in stroke patients.

Although the topic could be of interest, the study design and the presence of methodological issues may influence the results and may not support the conclusions. Without a control group, it could be hard to verify the effectiveness of this program on return to work.

The Authors stated that two occupational therapists were responsible for coordinating the return to work programme, but it is not crystal clear how the Authors reduced the risk of bias during the WRI and AWP assessments and during the data analysis.

The sample is very small, in addition, the patients have different types of disabilities and were included in the program at different times since the stroke. No information about the homogeneity of the sample at baseline was provided.

A detailed description of an intervention is required, also referring to the programme described in “Johansson U, Hellman T, Öst Nilsson A, Eriksson G. The ReWork-Stroke rehabilitation programme described by use of the TIDieR checklist. Scand J Occup Ther. 2020;1-9. doi:10.1080/11038128.2020.1790654”, in order to make the proposed programme replicable and implementable in future studies.

The Authors should better clarify the definition of “mild” and “moderate” stroke severity, and the explanation of the correlated abbreviation “BI”, present in the table 1.

The presentation of tables in the text is a bit confusing. To facilitate the reading, the authors should report in the correct order in which they appear in the text. Table 1 lacks description, some abbreviations and asterisks (*) in the last row are not explained. Please define the abbreviations ADL and IADL, both in the text and in table 2. Tables are sometimes indicated with a capital letter and sometimes with a lowercase letter. Can’t find in the text the reference number 42: “WMA: World Medical Association Declaration of Helsinki: ethical principles for medical research involving human subjects. 2013. Available from: https://www.wma.net/policies-post/wma-declaration-of helsinki”

In the light of the points above, in order to avoid misunderstandings in the journal’s readers, it could be useful modify the title, the aims and the conclusions in order to resubmit the study as a feasibility study that explores the possibility to introduce the ReWork-stroke programme in the rehabilitation of post-stroke patients.

Reviewer 2 Report

Overall, a well-written paper with some interesting information that can be useful to a variety of clinicians, especially when teasing out the details/domains of each measure used. There are a few minor grammatical errors throughout therefore suggest another review to correct those. Additionally, here are some additional comments/questions.

Lines 191-193: This paper stated that clinically meaningful change was determined by a change of +/- 15 points however the reference cited for this was based on the SIS version 2.0. This study used version 3.0 which has a different number of total items (64 items on version 2.0 versus 59 on 3.0). Studies have reported vastly different clinical meaningful change scores for version 3.0 (https://journals.sagepub.com/doi/pdf/10.1177/1545968309356295)

Lines 225-226: Reported the SIS domains that were ‘affected’ at baseline. What score was used to determine this? Anything below 100 or is there a particular cut-off?

For this work-rentry program, was the purpose to return each person to their original jobs? Or were new jobs given unrelated to their original work?

Table 3: It seems as though two participants had worsening work potential (I & J) which was not one of the three who were still in the work rentry program. Does this mean that work potential did not correlate with obtaining and maintaining a job?

Author Response

Response to comments from reviewer 2

  • Overall, a well-written paper with some interesting information that can be useful to a variety of clinicians, especially when teasing out the details/domains of each measure used. There are a few minor grammatical errors throughout therefore suggest another review to correct those. Additionally, here are some additional comments/questions. 

Response: The manuscript was professionally language edited before submission. However, we have now gone through the whole manuscript once again in order to condense the text and correct grammatical errors.

  • Lines 191-193: This paper stated that clinically meaningful change was determined by a change of +/- 15 points however the reference cited for this was based on the SIS version 2.0. This study used version 3.0 which has a different number of total items (64 items on version 2.0 versus 59 on 3.0). Studies have reported vastly different clinical meaningful change scores for version 3.0 (https://journals.sagepub.com/doi/pdf/10.1177/1545968309356295)

Response: It is correct that we are using SIS 3.0 and we are very aware about the study on SIS 3.0 that you refer to here. In that paper they calculated clinically important differences for four of the eight SIS domains; strength, hand function, activities of daily living/instrumental activities of daily living (ADL/IADL), and mobility. The sample Lin et al recruited for this study received specific motor training due to a paretic arm after stroke. We chose using clinically meaningful change determined based on Duncan et al 1999, “The Stroke Impact Scale Version 2.0 : Evaluation of Reliability, Validity, and sensitivity to change” for two reasons. The Lin et al study provides data on clinically important change only for four of the domains and we were interested in all of the domains and what might have been clinically meaningful also for the domains Emotion and Participation as those might be influenced by a return-to-work process and Memory and Communication as difficulties in these might be interfering with returning to work. Numbers of items differs between versions 2.0 and 3.0; some were removed after conducting the Rasch analysis. However, Duncan et al in her 1999 paper states that changes in SIS domain scores of approximately 10 to 15 points appear to represent reasonable definitions of clinically meaningful change. That definition has been used previously in numbers of SIS-studies and therefore by the choice of using a change of +/- 15 we can compare changes in impact of stroke over time with other samples receiving rehabilitation.

  • Lines 225-226: Reported the SIS domains that were ‘affected’ at baseline. What score was used to determine this? Anything below 100 or is there a particular cut-off?

Response: Thank you for this comment. We realize that this can be more thoroughly described. We have not used a specific cut off when we reported participants being affected. We looked at each person´s perceived impact at baseline and found that in the domains Emotion, Participation and in Stroke Recovery half or more of them had a calculated domain score at about 60 or below.

We have tried to clarify this in the text on page 6 (added text are indicated here with bold text): The domains of Emotions, Participation, and Stroke recovery were affected (domain score about 60) at baseline for nearly all participants and continued to be affected at follow-up.

  • For this work-rentry program, was the purpose to return each person to their original jobs? Or were new jobs given unrelated to their original work?

Response: The participants were included as they were working before stroke and wanted to return to their work. All of them had an employment and the main goal was to get back to that employment. All participants except one were at their workplaces during work trial. In that case the participant discussed with the employer and the OT that the work was too demanding for that person after stroke and the participant found a new workplace for the work trial. We have clarified the participants return to their previous jobs by adding text in the description of the ReWork-Stroke on page 3:

“A plan concerning a work trial and time for follow-up were elaborated at the client´s workplace in cooperation with the other stakeholders involved”

  • Table 3: It seems as though two participants had worsening work potential (I & J) which was not one of the three who were still in the work rentry program. Does this mean that work potential did not correlate with obtaining and maintaining a job?

Response: In this case work potential did not correlate with work re-entry for those participants. There are a lot of aspects, besides work potential that can be of importance in returning to work. However, the sample size is so small so we cannot argue for any associations between results on the assessments and RTW at all. Hopefully, any such association can be explored in a future larger sample.

Reviewer 3 Report

This study shows how OT program can help to return stroke victims to their work place. It presents 10 cases who used this RE-work Stroke program. I thin it is interesting approach and reading this will benefit to allied health professionals, social workers and stroke physicians, however this manuscript can be improved.

  1. All sections are wordy and can be shortened
  2. Conclusion is very generic. I think authors should mention what strokes benefit the most from their program? Right hemisphere, left hemisphere?
  3. It is understandable that only 10 patients included in the study, but at least a direction for generalization would be helpful. what are the future points?

Author Response

Response to reviewer 3

  • This study shows how OT program can help to return stroke victims to their work place. It presents 10 cases who used this RE-work Stroke program. I thin it is interesting approach and reading this will benefit to allied health professionals, social workers and stroke physicians, however this manuscript can be improved.

All sections are wordy and can be shortened

Response: Thank you for this comment. The manuscript was professionally language edited before submission however we have now gone through the whole manuscript once again in order to condense the language and shorten the sentences.

  • Conclusion is very generic. I think authors should mention what strokes benefit the most from their program? Right hemisphere, left hemisphere?

Response: This is a very interesting thought. However, with the design of this study and the data that are collected, we cannot say anything about the benefit of the program due to localization of the stroke. A reason for that is that there are so many aspects that can impact on return to work for people after stroke. Fundamental for the program is the person-centered intervention based on the needs of the specific person. 

  • It is understandable that only 10 patients included in the study, but at least a direction for generalization would be helpful. what are the future points?

Response: This study was the first study where we tried out and implemented the ReWork-Stroke in clinical practice to see whether it worked in practice or not. We cannot generalize that much from our findings from this study as the sample is so small. However, after the program 7 out of the 10 participants started paid work which is a result indicating the feasibility of using a coordinator facilitating the rehabilitation to work process for people with stroke. In order to address your comment we have added suggestions for future studies in the conclusion on page 16.

This is however the first study on the ReWork-Stroke programme and further studies are needed to evaluate changes in work potential and work performance and the programme´s effectiveness in increasing work re-entry for people who have had stroke.”

Round 2

Reviewer 1 Report

The authors have successfully addressed the points I raised. From my point of view the clarity of the manuscript has improved after revision. I have no further comments.